# Evaluation of Intensity- and Contour-Based Deformable Image Registration Accuracy in Pancreatic Cancer Patients

**DOI:** 10.3390/cancers11101447

**Published:** 2019-09-27

**Authors:** Yoshiki Kubota, Masahiko Okamoto, Yang Li, Shintaro Shiba, Shohei Okazaki, Shuichiro Komatsu, Makoto Sakai, Nobuteru Kubo, Tatsuya Ohno, Takashi Nakano

**Affiliations:** 1Gunma University Heavy Ion Medical Center, Maebashi 371, Japan; 2Graduate School of Medicine, Gunma University, Maebashi 371, Japan

**Keywords:** deformable image registration accuracy, pancreatic cancer, intensity- and contour-based deformable image registration, mean displacement to agreement

## Abstract

We aimed to clarify the accuracy of rigid image registration and deformable image registration (DIR) in carbon-ion radiotherapy (CIRT) for pancreatic cancer. Six patients with pancreatic cancer who were treated with passive irradiation CIRT were enrolled. Three registration patterns were evaluated: treatment planning computed tomography images (TPCT) to CT images acquired in the treatment room (IRCT) in the supine position, TPCT to IRCT in the prone position, and TPCT in the supine position to the prone position. After warping the contours of the original CT images to the destination CT images using deformation matrices from the registration, the warped delineated contours on the destination CT images were compared with the original ones using mean displacement to agreement (MDA). Four contours (clinical target volume (CTV), gross tumor volume (GTV), stomach, duodenum) and four registration algorithms (rigid image registration [RIR], intensity-based DIR [iDIR], contour-based DIR [cDIR], and a hybrid iDIR-cDIR ([hDIR]) were evaluated. The means ± standard deviation of the MDAs of all contours for RIR, iDIR, cDIR, and hDIR were 3.40 ± 3.30, 2.2 1± 2.48, 1.46 ± 1.49, and 1.46 ± 1.37 mm, respectively. There were significant differences between RIR and iDIR, and between RIR/iDIR and cDIR/hDIR. For the pancreatic cancer patient images, cDIR and hDIR had better accuracy than RIR and iDIR.

## 1. Introduction

As carbon-ion beams have the characteristics of a Bragg peak and sharper penumbra [1], they can generate a more conformal dose distribution than X-ray beams [2]. However, there is the risk that the dose distribution may be substantially changed if the target position moves or the internal organ contours around the target change. In particular, it was reported that changes were observed for mobile organs that moved with respiratory movement [3,4,5,6,7,8]. To safely treat such organs, it is necessary to confirm the reproducibility of the dose distributions. Additionally, confirming the dose distributions during the whole treatment days can obtain a more accurate estimation for the irradiated dose, and it is effective for predicting treatment outcomes and toxicities [9]. For dose distribution confirmation, it is necessary to acquire computed tomography (CT) images on whole treatment days, to calculate the dose distributions on the CT images, and to accumulate the overall dose distributions [7,9,10]. 

To calculate the accumulated dose, it is necessary to use deformable image registration (DIR) to calculate the deformation matrix representing the correspondence for each pixel between the two CT image sets. This can be problematical because the accuracy of DIR might be reduced when it is applied to intricately changing tissue, and the accuracy of DIR may be especially unreliable for the bowels, where there are a lot of gas-containing regions. Houweling et al. used a rigid image registration (RIR) technique for the registration between treatment CT images and other CT images in pancreatic cancer cases because they were concerned about uncertainties due to internal changes [8]. However, it is difficult to match a point in the CT image of deformed tissue with a corresponding point in another CT image when using RIR, and it is unknown whether the accuracy of RIR is better than that of deformable image registration (DIR). Carbon-ion radiotherapy (CIRT) has better outcomes than other therapies in pancreatic cancer, but nevertheless, the local control rate for locally-advanced pancreatic cancer is only 70%, which is not satisfactory [11,12]. Therefore, it is important to investigate the accuracy of DIR in pancreatic cancer patient images to ensure accurate evaluation of the accumulating dose. Additionally, it is also important to investigate the accuracy of DIR in various patient positions, such as supine and prone, because few facilities have a rolling gantry for CIRT [13] and various patient positions are used in CIRT for pancreatic cancer.

Several evaluations of the accuracy of DIR verifications have been reported [14,15,16]. These showed that DIR has good accuracy in lung, liver, or prostate cancer patient images with same patient positions. However, the accuracy in pancreatic cancer patient images, where the bowels can show substantial shape and position changes, has not been evaluated, and neither has the accuracy between different patient positions, such as supine and prone positions.

In this study, we aimed to clarify the accuracies of RIR and DIR when applied to prospectively acquired patient CT images for CIRT for pancreatic cancer.

## 2. Results

Sample CT images with transferred contours and dose distributions of supine and prone positions are shown in Figure 1 and Figure 2, respectively. The dice similarity coefficient (DSC) and mean displacement to agreement (MDA) for each contour for all patterns are shown in Figure 3. In all cases, the means ± standard deviation of the DSCs of all contours (*n* = 72) were 0.72 ± 0.20, 0.81 ± 0.16, 0.86 ± 0.11, and 0.87 ± 0.08 with RIR, intensity-based DIR (iDIR), contour-based DIR (cDIR), and hybrid iDIR-cDIR (hDIR), respectively; the means ± standard deviation of the MDAs were 3.40 ± 3.30, 2.21 ± 2.48, 1.46 ± 1.49, and 1.46 ± 1.37, respectively. There were significant differences between RIR and the other registration methods for both the DSCs and MDAs (*p* < 0.001 for all combinations). Moreover, the DSCs and MDAs of cDIR and hDIR were significantly better than those of RIR and iDIR (for both DSCs and MDAs, *p* < 0.001 for cDIR vs. RIR, and hDIR vs. RIR). For registration from the first irradiation day (1st-IRCT) to the supine position CT images for treatment planning (SP-PlanCT) (SP-SP) cases, the MDAs were less than 2 mm in 45.8% of RIR cases, 83.3% of iDIR cases, 100% of cDIR cases, and 95.8% of hDIR cases. The corresponding values for registration from the 10th-IRCT to the prone position CT images for treatment planning (PR-PlanCT) (PR-PR) cases were 54.2%, 79.2%, 95.8%, and 91.7%, respectively, while for registration from the PR-PlanCT to the SP-PlanCT (SP-PR) cases they were 37.5%, 33.3%, 54.2%, and 54.2%. Differences in the dose-volume parameters for all patterns are shown in Table 1. In all cases, the median differences for (clinical target volume (CTV) V95 were 2.47%, 1.65%, 2.24%, and 1.50% for RIR, iDIR, cDIR, and hDIR, respectively; for gross tumor volume (GTV) V95 the corresponding values were 0.84%, 0.81%, 2.47%, and 1.59%; for stomach V50 they were 1.60%, 2.94%, 0.85%, and 7.56%; and for duodenum V50 they were 0.68%, 0.91%, 0.58%, and 0.68%. Graphs showing the correlations between MDA and DSC, and between MDA and dose-volume parameters, are shown in Figure 4. The correlation coefficients *R* between MDA and DSC were 0.92, 0.93, 0.91, and 0.84 for RIR, iDIR, cDIR, and hDIR, respectively; the corresponding values between MDA and target (CTV and GTV) V95 were 0.47, 0.65, 0.56, and 0.40; and between MDA and organs at risk (OAR) (stomach and duodenum) V50 they were 0.01, 0.29, 0.23, and 0.44.

## 3. Discussion

The accuracy of DIR was better than that of RIR in cases with the same patient position, while the accuracies of cDIR and hDIR for the CTV, stomach, and duodenum were better than those of iDIR in all patient positions, as shown in Figure 3. Meanwhile, for the GTV, the accuracy of iDIR was better than the other methods. We assume that iDIR is effective when deformations are small and the boundary of the contour is clear. However, we assume that iDIR is not effective when the boundary of the contour is unclear (such as with the CTV), because obtaining pixel-by-pixel correspondence is difficult, which is also the case when CT values show large differences due to changes in gas content (such as in stomach and duodenum). Because cDIR and hDIR are less affected by the above effects, they were better than iDIR for the CTV, stomach, and duodenum. 

In previous studies, Motegi et al. reported a DSC of 0.96 ± 0.01 for the prostate with hDIR [15], and Sarudis et al. reported a DSC of more than 0.83 for the GTV with iDIR when the GTV was shifted more than 1.6 cm [16]. As the DSCs of GTV with hDIR and iDIR were 0.88 ± 0.07 and 0.89 ± 0.08, respectively, in this study, our accuracies were worse than those given in the above references. It is assumed that registration in pancreatic cancer patients, which have different patient positions, is difficult. Task Group (TG)-132 in the American Association of Physicists in Medicine (AAPM) guidelines shows that the tolerance of the MDA should be from 2 mm to 3 mm [17]. The MDA obtained with cDIR was the best in this study, being less than 2 mm in 100% of SP-SP cases and 95.8% of PR-PR cases. However, the value in SP-PR cases was much lower (54.2%). The average MDA of the target was approximately 1 mm, in contrast with the errors for the OARs (stomach and duodenum), which were approximately 2 mm. Large morphological changes occur in the bowels due to changes in their content and gases (stomach volume changed by a maximum of 158.97 ml and duodenal volume by 46.24 mL), and CT values can show large changes due to changes in gas content. This seems to be one cause of DIR failure. In particular, cDIR had MDA errors of 1.89 ± 2.27 mm in the stomach, and hDIR had errors of 1.51 ± 0.65 mm in the duodenum. In this study, we applied cDIR and hDIR to several contours (CTV, GTV, stomach, and duodenum) simultaneously; however, it is possible that accuracy would increase if each contour were used individually. Manual user configurations, such as Reg refine [18,19], or other new methods, are necessary to increase the accuracy of DIR, because the accuracy appears to be limited, even when cDIR and hDIR methods are used.

RIR and cDIR had approximately 10% maximum differences for CTV V95, and that all registrations had approximately 20% maximum differences for stomach V50, as shown in Table 1. The deformation of dose distribution in this study used only a simple deformation according to the deformation matrix, and it did not take into account preservation of the total dose with respect to the contour volumes and CT values. Thus, it is unknown whether the differences are reasonable or not. However, it should be noted that such a difference was observed when dose distributions transferred to other CT images were evaluated. Additionally, it should also be noted that a difference was observed when accumulating dose distributions across several CT image sets.

The correlation coefficients R between MDA and DSC were high (0.84–0.93), as shown in Figure 4. Although DSC accuracy generally increases as the volumes increases, DSC can be used for evaluating DIR accuracy in pancreatic cancer patient images because it has a high correlation with MDA. In contrast, the correlation coefficients between MDA and V95 for the target were only mid-range (0.40 to 0.65), because differences in the target V95 might decrease when registration accuracy increases. The correlation coefficients between MDA and V50 for the OARs were low (0.01, 0.29, and 0.23 with RIR, iDIR, and cDIR, respectively). It is not necessarily the case that the differences in the OAR V50 decrease when registration accuracy increases.

In the evaluation results using contours, cDIR and hDIR were better than RIR and iDIR. However, they can only be used by taking the correspondence pixel-by-pixel and accumulating the dose distribution, they cannot be used when transferring contours from one CT image set to another one; in such a case, iDIR might be more useful than RIR.

Our study has some limitations. We used only 18 (three sets for each of six patients) CT image sets, and further analysis is necessary to accurately evaluate DIR accuracy. Additionally, the registration evaluation used a relative evaluation based on contours. Even if the MDA was 0 (or DSC becomes 1), it is not necessarily the case that a point in the original CT image set matched the corresponding point in another CT image set. Therefore, it is possible that the evaluation underestimates the DIR error. Moreover, delineation errors are also included in this evaluation, even though the same oncologist delineated all contours.

## 4. Materials and Methods 

### 4.1. Patients

This prospective study included eight consecutive patients with pancreatic cancer who were each treated with 12 fractions of passive-irradiation CIRT at our facility between March 2018 and February 2019. This prospective study was performed to evaluate inter-fractional anatomical changes and to calculate the accumulated inter-fraction dose using daily CT images acquired in the treatment room, including determination and evaluation of the most appropriate method. Data from six patients were used in this study, with the data from two patients being excluded because they did not meet the following inclusion criteria: (i) CT images sets were acquired at treatment planning and other timepoints; (ii) CT image sets were acquired in both supine and prone positions; (iii) the patients had a stomach and duodenum. The patients’ characteristics are shown in Table 2. This study was conducted in accordance with the Declaration of Helsinki and was approved by our facility’s institutional review board (1564). The study was registered at the University Hospital Medical Information Network Clinical Trials Registry (UMIN-CTR trial number: 000029495). All patients gave written informed consent for inclusion before they participated in the study, and their data were anonymized.

### 4.2. CT Image Acquisition

Twelve CT data sets were acquired for each patient, one on each day of treatment, to investigate the effects of tumor movement and inter-fractional changes on the planned dose. The CT images for treatment planning (PlanCT) were acquired on a scanner in the simulation room (Aquilion LB^®^, Self-Propelled, Canon Medical Systems, Japan).

On each of the twelve separate radiotherapy days, the patient was irradiated after patient positioning was performed using orthogonal X-ray images [20], with supine positioning being used on days 1 to 9, and prone positioning for days 10 to 12. After the irradiation, a CT data set was acquired in the treatment room with the patient in the same position as that used for the irradiation, and with the same tube voltage, tube current, field of view, and slice thickness settings used for the PlanCT.

Four CT image sets were used in this study: PlanCT images with the patient in the supine position (SP-PlanCT), CT images acquired in the supine position in the treatment room on the first irradiation day (1st-IRCT), PlanCT images in the prone position (PR-PlanCT), and CT images acquired in the prone position in the treatment room on the 10th irradiation day (10th-IRCT). The median period (range) from the SP-PlanCT to the 1st-IRCT was 12 days (8–14 days), that to the PR-PlanCT was 15.5 days (9–20 days), and that to the 10th-IRCT was 27 days (22–29 days).

### 4.3. Treatment Planning and Dose Calculation

The Gunma University Heavy Ion Medical Center (GHMC) provides carbon-ion therapy [21] using a heavy ion irradiation device (Mitsubishi Electric, Japan) with a passive irradiation method [22]. The passive irradiation field was generated using a scatterer and wobbling, and the field was collimated to the outside of the planning target volume (PTV) using a multi-leaf collimator (MLC). A treatment planning system with a pencil-beam algorithm (XiO-N, Elekta Sweden, Mitsubishi Electric, Japan) was used. The relative biological effectiveness (RBE) was included in the absorbed dose using a spread-out Bragg peak concept [23], and the clinical dose, including this, was defined as Gy (RBE). 

A radiation oncologist delineated the stomach, duodenum, and intestine (bowels) on each PlanCT while referring to contrast-enhanced CT images. The gross tumor volume (GTV); clinical target volume-1 (CTV1; by adding 5-mm margins to the GTV, including the prophylactic lymph node, and excluding the bowels + 2 mm [excluding the GTV]), and CTV2 (by adding 5-mm margins to the GTV, excluding the bowels + 2 mm) were also delineated.

The dose distribution of the anterior–posterior (AP) beam field was calculated on the SP-PlanCT, and the dose distribution was calculated on the 1st-IRCT using the same parameters as used on the SP-PlanCT. The dose distribution of the posterior–anterior (PA) beam field was calculated on the PR-PlanCT, and also on the 10th-IRCT using the same parameters. The prescribed dose was set to 4.6 Gy (RBE). The priority for the AP beam field was to ensure target coverage, while the priority for the PA beam field was to spare organs at risk (OARs). Dose distributions on the 1st-IRCT and 10th-IRCT were analyzed.

### 4.4. Deformable Image Registration Algorithm

Four types of registration algorithm were used: rigid image registration (RIR), intensity-based deformable image registration (iDIR), contour-based deformable image registration (cDIR), and hybrid intensity- and contour-based deformable image registration (hDIR). The DIR algorithms were implemented by the VoxAlign Deformation Engine in MIM maestro (MIM Software Inc., USA). For the cDIR and hDIR, all the four contours delineated on each CT image were used: the CTV1 (or 2), GTV, stomach, and duodenum.

### 4.5. Data Analysis

Registration from the 1st-IRCT to the SP-PlanCT was defined as SP-SP, registration from the 10th-IRCT to the PR-PlanCT as PR-PR, and registration from the PR-PlanCT to the SP-PlanCT as SP-PR. In each case, the contours on one CT set were warped and transferred to the other CT set using the deformation matrix from each registration. The mean distance to agreement (MDA) and dice similarity coefficient (DSC) were calculated between the delineated contours on the CT images and transferred contours. The CTV1 (or 2), GTV, stomach, and duodenum, were used for the evaluations. 

Furthermore, in each case, the dose distribution on one CT set was warped and transferred to the other CT set using the deformation matrix from each registration. The dose-volume parameters were compared between the dose distribution on original CT set and the transferred dose distribution on the other CT set using contours delineated on each CT set, and the difference between each parameter was calculated. The CTV1 (or 2) and GTV receiving greater than 95% of the prescription dose (V95) and the stomach and duodenum V50 and V10 were used for the calculations. Additionally, for each case, the correlations between MDA and DSC, and between MDA and dose-volume parameter indices, were calculated. 

The Bonferroni method was used to correct for multiple comparisons in MDA or DSC measurements between RIR, iDIR, cDIR, and hDIR. A level of *p* < 0.05 was considered statistically significant. Statistical analyses were performed using SPSS software (IBM SPSS Statistics for Windows, version 25.0, IBM, Inc., Armonk, NY, USA).

## 5. Conclusions

In this study, we evaluated RIR and intensity- and contour-based DIR accuracy in CIRT for pancreatic cancer patients. We found that DIR accuracy was significantly better than RIR accuracy for inter-fractional CT image sets with the same patient position and that contour-based DIR and hybrid DIR were significantly better than RIR and intensity-based DIR for inter-fractional CT image sets with different patient positions.

## Figures and Tables

**Figure 1 cancers-11-01447-f001:**
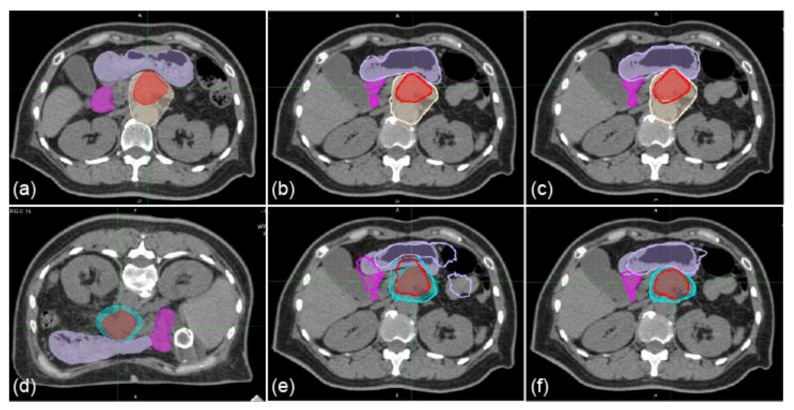
Sample axial images of supine and prone positions with delineated and transferred contours. (**a**) first irradiation day (1st-IRCT). (**b**,**c**) supine position CT images for treatment planning (SP-PlanCT) images. (**d**) 10th-IRCT images. (**e**,**f**) prone position (PR)-PlanCT images. Filled red, beige, cyan, mauve, and magenta regions show delineations of the gross tumor volume (GTV), clinical target volume 1 (CTV1), CTV2, stomach, and duodenum, respectively. Bold red, beige, mauve, and magenta regions show the GTV, CTV1, stomach, and duodenum transferred from the 1st-IRCT to SP-PlanCT images with intensity-based deformable image registration (iDIR) (**b**) and contour-based DIR (cDIR) (**c**). Bold red, cyan, mauve, and magenta regions show GTV, CTV2, stomach, and duodenum transferred from PR-PlanCT images to SP-PlanCT images with iDIR (**e**) and cDIR (**f**).

**Figure 2 cancers-11-01447-f002:**
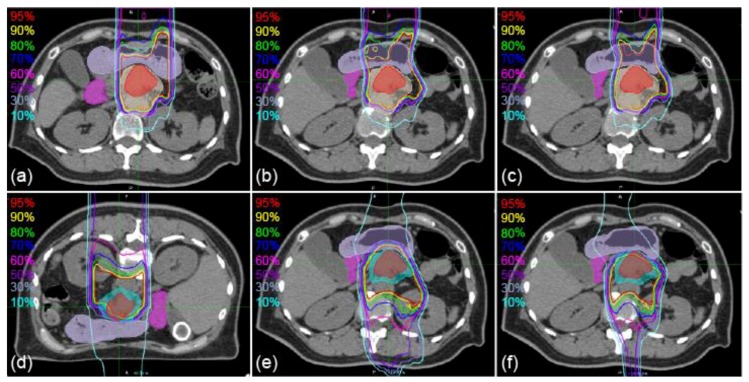
Sample axial images of supine and prone positions with the dose distributions of vertical beam fields. (**a**) 1st-IRCT images. (**b**,**c**) SP-PlanCT images. (**d**) 10th-IRCT images. (**e**,**f**) PR-PlanCT images. Filled red, beige, cyan, mauve, and magenta show delineations of GTV, CTV1, CTV2, stomach, and duodenum, respectively. Dose distributions are transferred from the 1st-IRCT images (**a**) to SP-PlanCT images with iDIR (**b**) and cDIR (**c**). Dose distributions are transferred from PR-PlanCT images (**d**) to SP-PlanCT images with iDIR (**e**) and cDIR (**f**).

**Figure 3 cancers-11-01447-f003:**
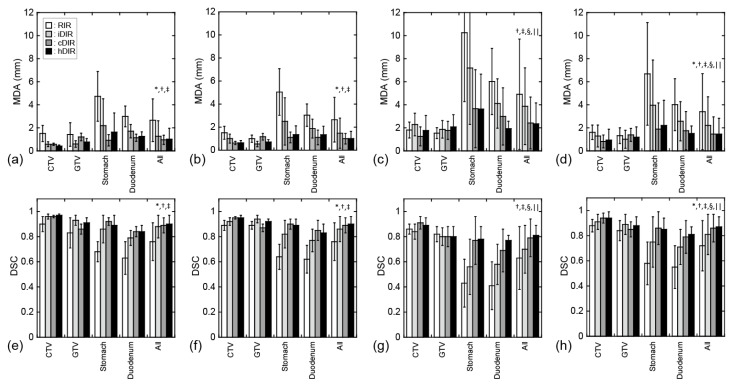
Registration errors using contours. All graphs show mean ± standard deviation. White, light-gray, dark-gray, and black bars show rigid image registration (RIR), iDIR, cDIR, and hybrid iDIR-cDIR (hDIR), respectively. (**a**–**d**) show the MDA for each contour, and (**e**–**h**) show the dice similarity coefficient (DSC) for each contour. (**a**,**e**) SP-SP cases. (**b**,**f**) SP-PR cases. (**c**,**g**) SP-PR cases. (**d**,**h**) all cases. *, †, ‡, §, and || indicate significant differences between RIR and iDIR, RIR and cDIR, RIR and hDIR, iDIR and cDIR, and iDIR and hDIR, respectively.

**Figure 4 cancers-11-01447-f004:**
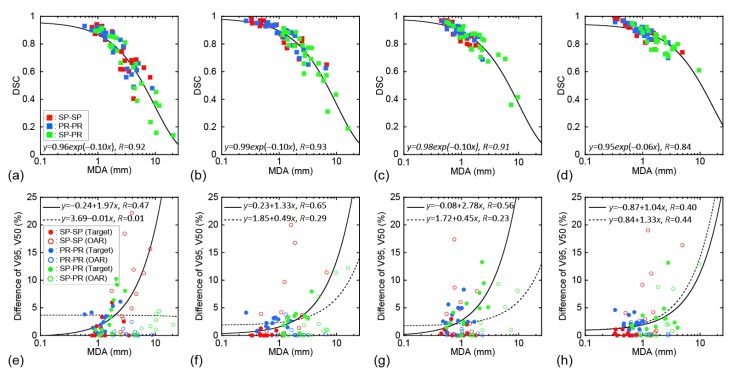
Correlations between mean displacement to agreement (MDA) and DSC (**a**–**d**), and MDA and absolute difference of V95 of CTV and GTV or V50 of stomach and duodenum (**e**–**h**). Red markers show the results of SP-SP cases, blue markers PR-PR cases, and green markers SP-PR cases. Squares represent target (CTV and GTV) and organs at risk (OARs) (stomach and duodenum), filled circles represent target, and hollow circles OAR. (**a**,**e**) show RIR, (**b**,**f**) show iDIR, (**c**,**g**) show cDIR, and (**d**,**h**) show hDIR. The black line in (**a**–**d**) indicates a regression curve fitted with an exponential function. The black and black-dotted lines in (**e**–**h**) show regression lines fitted with a linear function for the target and OAR, respectively.

**Table 1 cancers-11-01447-t001:** Absolute differences in the dose-volume parameters. All data show the median (range).

Case	Volume	Parameter	RIR	iDIR	cDIR	hDIR
SP-SP	CTV	V95 (%)	2.10 (0–5.94)	0.06 (0–1.28)	0.74 (0–2.97)	0.36 (0–1.10)
	GTV	V95 (%)	0.00 (0–0.67)	0.00 (0–0.28)	0.00 (0–1.98)	0.00 (0–0.93)
	Stomach	V50 (%)	9.38 (2.99–22.18)	6.79 (0.84–19.99)	6.39 (0.18–17.38)	6.68 (1.15–19.01)
		V10 (%)	13.77 (7.38–23.54)	14.33 (6.88–22.22)	14.51 (6.90–22.49)	13.29 (6.74–24.60)
	Duodenum	V50 (%)	6.49 (0.24–18.42)	3.39 (0.47–16.76)	1.43 (0.10–11.85)	1.77 (0.30–22.84)
		V10 (%)	5.48 (0.04–19.78)	3.41 (0.69–28.10)	4.71 (0.10–11.85)	5.71 (0.30–22.84)
PR-PR	CTV	V95 (%)	2.95 (0.40–6.12)	2.40 (1.34–3.24)	2.60 (0.57–5.61)	2.40 (1.79–4.21)
	GTV	V95 (%)	2.59 (0.69–4.15)	1.95 (0.76–4.13)	3.71 (0.19–8.30)	2.10 (0.98–4.62)
	Stomach	V50 (%)	0.11 (0.00–1.22)	0.15 (0.04–3.81)	0.25 (0.00–2.32)	0.23 (0.07–2.54)
		V10 (%)	2.57 (1.22–7.45)	1.41 (0.35–9.50)	1.72 (1.11–10.81)	1.05 (0.70–11.26)
	Duodenum	V50 (%)	0.22 (0.00–2.45)	0.04 (0.00–0.98)	0.12 (0.00–2.15)	0.24 (0.00–2.32)
		V10 (%)	0.78 (0.21–15.28)	1.41 (0.18–4.82)	1.56 (0.37–11.50)	2.21 (0.73–7.34)
SP-PR	CTV	V95 (%)	4.15 (0.15–10.20)	3.18 (0.42–7.95)	3.17 (1.43–9.46)	1.50 (1.15–4.86)
	GTV	V95 (%)	2.15 (0.00–9.16)	3.02 (0.00–3.57)	5.01 (0.00–13.25)	3.11 (0.00–13.15)
	Stomach	V50 (%)	1.51 (0.02–4.40)	4.01 (0.00–12.15)	0.51 (0.00–8.05)	1.97 (0.00–12.91)
		V10 (%)	6.97 (0.31–21.37)	7.96 (0.28–13.51)	5.81 (0.25–30.05)	7.56 (0.01–11.34)
	Duodenum	V50 (%)	0.82 (0.08–3.93)	0.91 (0.02–2.09)	0.53 (0.00–9.30)	0.98 (0.00–8.77)
		V10 (%)	3.96 (1.74–22.19)	5.10 (0.12–30.80)	3.31 (2.02–19.89)	4.83 (0.81–17.98)
All	CTV	V95 (%)	2.47 (0–10.20)	1.65 (0–7.95)	2.24 (0–9.46)	1.50 (0–4.86)
	GTV	V95 (%)	0.84 (0–9.16)	0.81 (0–4.13)	2.47 (0–13.25)	1.59 (0–13.15)
	Stomach	V50 (%)	1.60 (0.00–22.18)	2.94 (0.00–19.99)	0.85 (0.00–17.38)	2.46 (0.00–19.01)
		V10 (%)	7.42 (0.31–23.54)	8.19 (0.28–22.22)	8.40 (0.25–30.05)	7.56 (0.01–24.60)
	Duodenum	V50 (%)	0.68 (0.00–18.42)	0.91 (0.00–16.76)	0.58 (0.00–9.30)	0.68 (0.00–11.20)
		V10 (%)	2.10 (0.04–22.19)	2.06 (0.12–30.80)	2.48 (0.10–19.89)	3.78 (0.30–22.84)

RIR, rigid image registration; iDIR, intensity-based deformable image registration; cDIR, contour-based deformable image registration; hDIR, hybrid deformable image registration; CTV, clinical target volume; GTV, gross tumor volume. Vx, volume receiving greater than x% of the prescription dose.

**Table 2 cancers-11-01447-t002:** Patient characteristics. Each volume is represented by the mean ± standard deviation of the 4 CT sets.

Patient	Sex	Age	Patient Position	Tumor Position	Tumor Volume (mL)	Stomachic Volume (mL)	Duodenal Volume (mL)
1	F	50	SP0, PR0	Body	22.4 ± 3.3	238.3 ± 79.3	48.9 ± 6.3
2	F	84	SP0, PR10	Head	35.3 ± 2.6	185.1 ± 46.6	100.4 ± 19.0
3	M	82	SP0, PR350	Body	30.8 ± 2.1	227.0 ± 47.4	63.3 ± 14.3
4	F	61	SP0, PR0	Body	26.0 ± 1.7	185.1 ± 31.1	48.0 ± 1.5
5	F	77	SP0, RP350	Body	20.8 ± 2.6	151.7 ± 20.3	54.0 ± 7.9
6	M	74	SP0, PR0	Body	40.2 ± 3.4	153.5 ± 25.7	77.8 ± 15.3
Median	-	75.5	-	-	28.4	185.1	58.6

M, male; F, female; SP0, supine 0-degree position; PRx, prone x-degree position; Body, pancreatic body; Head, pancreatic head.

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
