# Peer review of "Evaluation of Intensity- and Contour-Based Deformable Image Registration Accuracy in Pancreatic Cancer Patients"

_cancers, 2019, doi:10.3390/cancers11101447_

Round 1
Reviewer 1 Report
Kubota et al presents a highly informative study evaluating the accuracy of rigid and deformable image registration of planning CT scans and same-day treatment CT scans. The study addresses an important issue related to accuracy of carbon-ion therapy of pancreatic therapy where the propensity for organ shift is high. The study provides an important contribution that may potentially improve radiotherapy outcomes for pancreatic cancer. The manuscript is very well-written and organized and concisely conveys the purpose and results of the evaluation.
Minor points:
The study offers a descriptive evaluation and lacks a mechanistic explanation of outcomes. While the poor accuracy of RIR in the context of soft tissue registration is intuitive, the reasons for the superior accuracy of cDIR and hDIR over iDIR are not as obvious. The reader may benefit from an analytical perspective in the Discussion section describing, or at least speculating, on the rationale for the differing outcomes of DIR methods. Line 135 mentions “dDIR and hDIR.” Is it intended to read as “cDIR and hDIR”?Author Response
The study offers a descriptive evaluation and lacks a mechanistic explanation of outcomes. While the poor accuracy of RIR in the context of soft tissue registration is intuitive, the reasons for the superior accuracy of cDIR and hDIR over iDIR are not as obvious. The reader may benefit from an analytical perspective in the Discussion section describing, or at least speculating, on the rationale for the differing outcomes of DIR methods.
> Thank you for your review and comments. While the accuracy of iDIR was better than the other methods for the GTV, it was less effective than cDIR and hDIR for the CTV, stomach, and duodenum. We assume that iDIR is effective when deformations are small and the boundary of the contour is clear. However, we assume that iDIR is less effective when the boundary of the contour is unclear (such as with the CTV), because obtaining a pixel-by-pixel correspondence is difficult. This is also the case when CT values show large differences due to changes in gas content, such as in the stomach and duodenum. Because cDIR and hDIR are less affected by the above effects, they are better than iDIR for the CTV, stomach, and duodenum.
We have added this content in revised Discussion section (128-136).
Line 135 mentions “dDIR and hDIR.” Is it intended to read as “cDIR and hDIR”?
>We have changed “dDIR” to “cDIR”.
Reviewer 2 Report
Review for Manuscript cancers-597293-peer-review-v1
General Comments: Very nicely written manuscript and extremely easy to review. First, a few general comments:
In the results section, P values are needed in the text to denote significant or non-significant differences. Also in the results section, there should be a greater explanation of the results rather than referencing the Figures and Tables. This is particularly the case for Table 1 and Figure 4. In the discussion, reword the first sentences of the paragraphs so that the sentences do not start with “Figure 3” for example.More Specific Comments:
Title – None
Abstract – None
Introduction – None
Results – See above
Discussion – See above and
Line 126 – Remove the comma after “stomach”Materials and Methods
Line 182 – Remove the extra letter/number before “Twelve” Line 240 – State the type of ANOVA used prior to the Bonferroni post-hoc analysis.Conclusions – None
Figures, Tables, and Legends
For Table 1, indicate which differences were significant For Figure 3 – Symbols indicating significant differences need explanations in the legend.Author Response
In the results section, P values are needed in the text to denote significant or non-significant differences. Also in the results section, there should be a greater explanation of the results rather than referencing the Figures and Tables. This is particularly the case for Table 1 and Figure 4.
>Thank you for your review and comments. We have added “In all cases, the means ± standard deviation of the DSCs of all contours (n = 72) were 0.72±0.20, 0.81±0.16, 0.86±0.11, and 0.87±0.08 in RIR, iDIR, cDIR, and hDIR, respectively; the means ± standard deviation of the MDAs were 3.40±3.30, 2.21±2.48, 1.46±1.49, and 1.46±1.37, respectively. There were significantly differences between RIR and the other registration methods for both of the DSCs and MDAs (p<0.001 in all combinations)”, “(for both of the DSCs and MDAs, p<0.001 between cDIR and RIR, or hDIR and RIR).” making reference to Figure 3.
We have also added “In all cases, the median differences for CTV V95 were 2.47%, 1.65%, 2.24%, and 1.50% for RIR, iDIR, cDIR, and hDIR, respectively; for GTV V95 the corresponding values were 0.84%, 0.81%, 2.47%, and 1.59%; for stomach V50 they were 1.60%, 2.94%, 0.85%, and 7.56%; and for duodenum V50 they were 0.68%, 0.91%, 0.58%, and 0.68%” making reference to Table 1, and “The correlation coefficients R between MDA and DSC were 0.92, 0.93, 0.91, and 0.84 for RIR, iDIR, cDIR, and hDIR, respectively; the corresponding values between MDA and target (CTV and GTV) V95 were 0.47, 0.65, 0.56, and 0.40; and between MDA and OAR (stomach and duodenum) V50 they were 0.01, 0.29, 0.23, and 0.44..” making reference to Figure 4.
In the discussion, reword the first sentences of the paragraphs so that the sentences do not start with “Figure 3” for example.
>We have changed the text for “Figure. 3 shows”, “Table 1 shows”, and “Figure. 4 shows”.
Line 126 – Remove the comma after “stomach”
>We have deleted the comma.
Materials and Methods
Line 182 – Remove the extra letter/number before “Twelve”
>We have deleted the extra letter/number.
Line 240 – State the type of ANOVA used prior to the Bonferroni post-hoc analysis.
> ANOVA can be generally omitted when using the Bonferroni, Dunnet, or Tukey-Kramer methods, because the difference between two specific groups may be missed. Therefore, we did not use ANOVA.
Figures, Tables, and Legends
For Table 1, indicate which differences were significant
>We did not test for significant differences between the values in Table 1, because the purpose was to check the reference differences in each method, not to evaluate accuracy.
For Figure 3 – Symbols indicating significant differences need explanation in the legend.
>We have added “*, †, ‡, §, and || indicate significant differences between RIR and iDIR, RIR and cDIR, RIR and hDIR, iDIR and cDIR, and iDIR and hDIR, respectively.”.
Reviewer 3 Report
Good and interesting paper
Congratulations
Author Response
Thank you for your review and comments. The revised manuscript has been edited by an English language editing service (Edanz Group Japan).
Round 2
Reviewer 2 Report
Review for Manuscript cancers-597293-peer-review-v2
General Comments: Thank you for addressing my comments/edits. In addition, thank you for clarifying that you performed the Bonferroni correction independent of a one-way ANOVA, which is a valid approach.